

# No effect of attentional modulation by spatial cueing in a masked numerical priming paradigm using continuous flash suppression (CFS)

Juliane Handschack[1], Marcus Rothkirch[1], Philipp Sterzer[1] and Guido Hesselmann[2]

[1] Charité—Universitätsmedizin Berlin, corporate member of Freie Universität Berlin and Humboldt-Universität zu Berlin, Department of Psychiatry and Neurosciences, Visual Perception Laboratory, Berlin, Germany
[2] Department of General and Biological Psychology, Psychologische Hochschule Berlin (PHB), Berlin, Germany

Corresponding author
Guido Hesselmann,
g.hesselmann@gmail.com

## ABSTRACT

One notion emerging from studies on unconscious visual processing is that different "blinding techniques" seem to suppress the conscious perception of stimuli at different levels of the neurocognitive architecture. However, even when only the results from a single suppression method are compared, the picture of the scope and limits of unconscious visual processing remains strikingly heterogeneous, as in the case of continuous flash suppression (CFS). To resolve this issue, it has been suggested that high-level semantic processing under CFS is facilitated whenever interocular suppression is attenuated by the removal of visuospatial attention. In this behavioral study, we aimed to further investigate this "CFS-attenuation-by-inattention" hypothesis in a numerical priming study using spatial cueing. Participants performed a number comparison task on a visible target number ("compare number to five"). Prime-target pairs were either congruent (both numbers smaller, or both larger than five) or incongruent. Based on the "CFS-attenuation-by-inattention" hypothesis, we predicted that reaction times (RTs) for congruent prime-target pairs should be faster than for incongruent ones, but only when the prime was presented at the uncued location. In the invisible condition, we observed no priming effects and thus no evidence in support of the "CFS-attenuation-by-inattention" hypothesis. In the visible condition, we found an inverse effect of prime-target congruency. Our results agree with the notion that the representation of CF-suppressed stimuli is fractionated, and limited to their basic, elemental features, thus precluding semantic processing.

## INTRODUCTION

Studies investigating the scope and limits of unconscious visual perception (*i.e.,* stimulus-related cognitive processing in the absence of stimulus visibility) vary with respect to how visibility is manipulated, a choice which may significantly affect a study's outcome and conclusions (*Rothkirch & Hesselmann, 2017*). One emerging view is that not all invisible

stimuli are equal, in the sense that different "blinding techniques" suppress the conscious perception of stimuli at different levels of the neurocognitive architecture (*Breitmeyer, 2015*; *Dubois & Faivre, 2014*). However, results on unconscious processing are often remarkably heterogeneous even for one particular suppression method, suggesting that further choices in experimental design and data analysis exert a strong influence on a study's outcome as well (*e.g.*, the assessment of stimulus visibility, or the exclusion of participants and trials based on residual stimulus awareness).

In recent years, the method associated with arguably the largest heterogeneity regarding the scope and limits of unconscious processing has been continuous flash suppression (CFS), a variant of interocular suppression (*Pournaghdali & Schwartz, 2020*; *Sterzer et al., 2014*; *Yang et al., 2014*). CFS renders stimuli invisible by flashing high contrast masks to one eye (usually to the observer's dominant eye), while a low contrast image is presented to the other eye (*Tsuchiya & Koch, 2005*). Following its introduction, CFS soon became widely used in studies on unconscious processing, because the early results seemed to suggest that this method (a) readily provides robust and successful suppression of almost any kind of visual stimulus, (b) allows for more complex unconscious processing due to long suppression durations of up to many seconds, (c) leaves especially action-related visuomotor processing—potentially mediated by the dorsal visual pathway—intact (*Hesselmann et al., 2018*). This popularity of CFS quickly resulted in a large and heterogeneous body of findings. Given these conflicting results, one fundamental question turned out to be whether visual processing under CFS is limited to low-level features, *e.g.*, stimulus shape, or reaches up to high-level qualities like semantics, *e.g.*, word or symbol meaning (*Moors et al., 2019*).

In a seminal study, *Kang, Blake & Woodman (2011)* presented semantically related and unrelated words, and analyzed the N400 component of the event-related potential (ERP) as a measure of semantic processing. The results showed no N400 effect (*i.e.*, comparing related word pairs *versus* unrelated word pairs) when stimuli were rendered invisible by CFS, and the authors concluded that semantic analysis does not occur in the absence of awareness induced by interocular suppression. While this conclusion is in good agreement with findings from binocular rivalry (*Blake, 1988*; *Zimba & Blake, 1983*), and fits well with current views on CFS (*Moors et al., 2017*), it seems reasonable to ask whether there is a yet to be determined parameter differentiating between experimental designs, given that some stimulus-driven effects observed under CFS may be due to the encoding of high-level, semantic information (for a review, *Lin & He (2009)*). In a more recent ERP study, the allocation of spatial attention was manipulated using a cueing paradigm in which target words appeared either at a cued or non-cued location (*Eo et al., 2016*). Surprisingly, the results showed semantic processing (as indexed by the N400 effect) when stimuli were not attended, but no unconscious semantic processing in the attended condition. *Eo et al. (2016)* proposed that spatial inattention attenuates the depth of interocular suppression, thus facilitating semantic stimulus processing at the unattended location. Based on their review of the literature, they also claimed that in CFS studies showing unconscious semantic processing, participants were often unable to predict the location of a suppressed stimulus (*e.g.*, because the stimulus was randomly presented either above

or below the fixation point). Location uncertainty of the suppressed stimulus may have prevented attention from fully operating in the location of the stimulus during CFS, resulting in semantic stimulus processing. In the following, we will refer to this working model as the "CFS-attenuation-by-inattention" hypothesis. The hypothesized mechanism may seem rather counterintuitive when considering that attention typically facilitates information processing (*Carrasco, 2011*). Previous research, however, has shown that (a) reduced attention attenuates the strength of interocular suppression, (b) semantic information such as word meaning can be processed without attention, lending some a priori plausibility to the mechanism (*Brascamp & Blake, 2012*; *Luck, Vogel & Shapiro, 1996*).

In two previous studies, we have tested the "CFS-attenuation-by-inattention" hypothesis. First, using functional magnetic resonance imaging (fMRI) and multivariate pattern analysis (MVPA), we investigated whether the decodability of object category (faces *versus* houses) increases when attention is diverted away from the CF-suppressed stimulus in a spatial cueing task (*Handschack et al., 2022*). In line with earlier studies (*Guggenmos et al., 2015*), decoding accuracies were significantly larger in the attended compared to the unattended condition for visible stimuli, but the MVPA results from the CFS conditions provided no support for the "CFS-attenuation-by-inattention" hypothesis. The aim of a second, behavioral study (co-authored by one of us) was to examine whether semantic priming effects increase when CF-suppressed numerical primes are presented at an unpredictable location (*Benthien & Hesselmann, 2021*). Specifically, participants performed a number comparison task on a visible target number ("compare number to five"). Prime-target pairs were either congruent (both numbers smaller, or both larger than five) or incongruent. Based on the "CFS-attenuation-by-inattention" hypothesis, we predicted that reaction times (RTs) for congruent prime-target pairs should be faster than for incongruent ones, but only when the prime location was uncertain (or, unpredictable). The RT data, however, did not provide evidence for the effect of location uncertainty on unconscious semantic processing under CFS. The results of an exploratory analysis suggested a "response conflict": whenever the prime-target sequence was ascending (*e.g.*, 1–8), or when the prime-target sequence was descending (*e.g.*, 9–2) the response in the opposite direction in the number comparison task (*i.e.*, "smaller than five", or "larger than five") was experienced as more effortful by participants, and was associated with longer RTs.

In this study, our aim was to go one step further and test the "CFS-attenuation-by-inattention" hypothesis in a spatial cueing paradigm using arrow cues, similar to the original study by *Eo et al. (2016)*. As in our previous behavioural study, we investigated numerical priming effects in a number comparison task. Based on the hypothesized mechanism, we tested the following prediction: If diverted visuospatial attention attenuates interocular suppression and facilitates the semantic processing of the unattended prime stimulus, then there should be a larger priming effect for primes presented at an uncued location than for primes presented at a cued location.

[1] Statistical power is influenced by both the number of participants and the number of trials per participant and condition, *i.e.,* measurement precision (*Baker et al., 2021*). For a method on how to control measurement precision at the level of individual participants and stimulus conditions, we refer the reader to *Biafora & Schmidt (2020)*. In our study, there were 80 trials per participant and condition of interest, in good agreement with general recommendations for response priming (*Schmidt, Haberkamp & Schmidt, 2011*).

[2] With both eyes open, participants viewed an object through a three cm wide hole in a DIN-A4-sized card, held at arm's length. While continuing to keep focus on the object, keeping the object centered in the hole, and with both eyes open, participants were instructed to slowly bring the card towards themselves until it touches their face. The eye over which participants had the test card centered was defined at the dominant eye. The test was repeated to verify the result. Eye dominance indexed in this way is referred to as sighting dominance (*Yang, Blake & McDonald, 2010*).

## MATERIALS & METHODS

### Participants

We report how we determined our sample size, all data exclusions, all inclusion/exclusion criteria, whether inclusion/exclusion criteria were established prior to data analysis, all manipulations, and all measures in the study. No part of the study procedures or analyses was pre-registered prior to the research being conducted. Please note that the data were collected earlier than the data from a previously published numerical priming study from our lab (*Benthien & Hesselmann, 2021*), but the data of the current study were analysed later. Some methodological details follow those used in our previous CFS studies (*Benthien & Hesselmann, 2021*; *Handschack et al., 2022*; *Rothkirch & Hesselmann, 2018*).

Twenty-nine participants took part in this study (mean age: 24 years, range: 18–42; 23 female, right handed: 28, dominant right eye: 27), and were recruited *via* student mailing lists. The final sample consisted of $N = 25$ participants (for details on data exclusions, see paragraph "Behavioural data analysis"). Sample size was determined based on a previous study which reported a target-specific effect of CF-suppressed numerical primes on RTs (*Hesselmann et al., 2015*). The reported partial eta-squared ($\eta_p^2$) for this effect (Exp.2) was 0.30. Using GPower 3.1.9.7 (*Faul et al., 2007*) we first converted $\eta_p^2$ to the effect size f(U), using the "SPSS effect size specification" option. We then determined that for an effect size $f(U) = 0.65$ and a power of 0.85 a sample of $N = 24$ was necessary (ANOVA repeated measures, within factor; alpha = 0.05; number of groups = 1; number of measurements = 2; epsilon = 1).[1]

All participants had normal or corrected-to-normal vision and no history of neurological or psychiatric disorders. All participants were naïve to the purpose of the experiment, provided written informed consent and received monetary compensation (€8/h) for their participation. Eye dominance was determined by the (distance) hole-in-card test (*Miles, 1930*).[2] This study was conducted at the Department of Psychiatry and Neurosciences, Charité—Universitätsmedizin Berlin, Germany, and approved by the Ethical Committee of the German Association of Psychology (Deutsche Gesellschaft für Psychologie, DGPs; Ethical Application Ref: GA_Hesselmann-092010-rev). The total duration of the experiment, including staircase, main experiment and two control experiments, was approximately 90 min.

### Setup and stimuli

The experiment was conducted in a darkened room, with indirect light coming from the experimenter's PC screen. Participants viewed a 17″ CRT monitor (SAMTRON 98PDF; effective screen diagonal: 43.6 cm; 1,280 × 960 pixels; refresh rate 60 Hz) through a mirror stereoscope using a chinrest to stabilize head position (Fig. S1). The viewing distance was 47 cm. Participants provided their responses *via* button press on a PC keyboard. All stimuli were created using Matlab 7.9.0 (MathWorks, Natick, MA, USA), the Psychophysics toolbox 3.0.12 (*Brainard, 1997*), and a ATI FireGL V7100 graphics card.

On a dark grey background, a medium grey rectangle was presented to each eye of the participant, framed by black and white stripes to aid binocular fusion (inner rectangle dimensions: 15.92° × 6.21°; outer dimensions: 16.11° × 6.43°; Fig. 1). At the center of

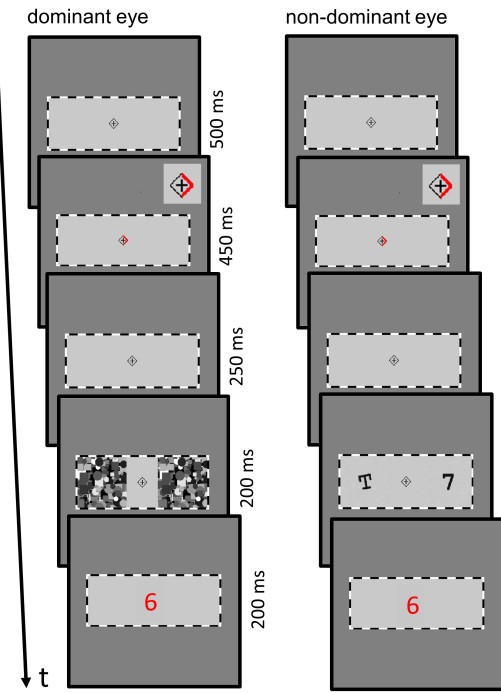

**Figure 1 Experimental paradigm.** Participants fixated the central fixation cross throughout the trial, and allocated their spatial attention to the left or right side as indicated by the red cue (Note that inset in the top right corner shows an enlarged version of the cue; for illustration purposes only). A black number prime and letter stimulus were presented simultaneously to the non-dominant eye, one stimulus at the attended location and the other stimulus at the unattended location. Stimuli were presented either in an upright or a tilted orientation. CFS masks were shown to the dominant eye to render number and letter stimuli invisible. In the visible condition (not shown), the stimuli were superimposed onto the CFS masks. Following the presentation of the letter/number stimulus, a central red target number appeared (slightly enlarged in the figure, for illustration purposes only). Participants were instructed to indicate as quickly and accurately as possible whether it was smaller or larger than five. In one block of trials, the participants' second task was to report the orientation of the attended stimulus. In another block of trials, participants provided a subjective visibility rating of the attended stimulus.

each rectangle, a black fixation cross was presented within a black rhombus. Participants were instructed to maintain fixation throughout each trial. In CFS trials ("invisible" trials), greyscale Mondrian-like masks were flashed to the dominant eye with a frequency of 10 Hz, while a digit and a capital letter were presented to the non-dominant eye. Visual contrasts of CF-suppressed digit/letter stimuli were determined for each participant individually (see below). We created 25 CFS masks with random circles and squares (each element covering approx. 4% to 18% of the mask area; 6.21° × 6.21°). In trials without CF-suppression ("visible" trials), digit and letter were superimposed onto the CFS masks. CFS masks were presented with an offset of 1.78° from the fixation cross. As the digit and letter stimuli were smaller than the CFS masks, their offsets from the fixation cross were slightly larger.

### *Main experiment*

At the beginning of each trial, a blank fixation screen was shown for 500 ms (Fig. 1). Next, a red arrow cue appeared on the left or right side of the central rhombus for 450 ms,

[3]In the study by *Eo et al. (2016)*, cue validity was 67% with respect to the location of the target word. The target word was presented in the cued location in the "congruent" (CFS) and "visible" (no-CFS) conditions, while it was presented in the uncued location in the "incongruent" (CFS) condition.

indicating the direction that covert attention needed to be shifted to. After another blank fixation screen for 250 ms, a single digit (1, 3, 7, or 9; *i.e.,* the priming stimulus) and a capital letter (K, X, T, N, or F) were presented at the individually determined visual contrast to the non-dominant eye for 200 ms. Digit and letter stimulus were presented in an upright orientation or with a rotation angle of 10° degrees counter-clockwise (necessary for the second task, see below). In the "invisible" trials (CFS condition), two CFS masks were flashed to the dominant eye at 10 Hz. In the "visible" trials (no-CFS condition), CFS masks were flashed to the dominant eye as well, and digit and letter stimuli at full visual contrast were additionally superimposed onto the CFS masks. Subsequently, with a stimulus onset asynchrony (SOA) of 200 ms, a red target number (2, 4, 6 or 8) was displayed binocularly at the centre of each eye's stimulus rectangle for 200 ms. Target stimuli were always presented at full visual contrast. Participants' first task was to report whether the target number was smaller or larger than five as quickly and accurately as possible, using the left (<5) or right (>5) arrow key. There was no time limit for the response.

The main experiment consisted of two blocks, which differed only with respect to the unspeeded second task. The aim of the second task was to foster attentional orienting to the cued position, irrespective of whether it contained the digit or the letter stimulus; at the same time, the second task served to estimate the visibility of the digit (prime) and letter stimuli. In one block, the second task required participants to decide whether the attended stimulus (digit or letter) was tilted or not. Both options ("upright", "tilted") were written above each other on the screen. In the other block, participants provided visibility ratings of the attended stimulus (digit or letter), using the perceptual awareness scale (PAS): "no experience" (PAS = 1), "weak experience" (PAS = 2), "almost clear experience" (PAS = 3) or "clear experience" (PAS = 4). The four options were displayed vertically on the screen. In both blocks, participants selected their answer using the arrow keys (up, down). By pressing the space bar participants confirmed their selection and started the next trial. On each trial, one option was pre-selected randomly by the stimulation script. The order of blocks was counter-balanced across participants.

Each block consisted of 320 trials in random order. There were four different prime digits, and four different target digits. Prime digits were presented either monocularly and together with CFS masks shown to the other eye, or were additionally superimposed onto the CFS masks (factor "visibility level", 2 levels). The prime digits were either presented at the attended/cued or at the unattended/uncued location (factor "attention", 2 levels). Thus, cue validity was 50% with respect to the location of the digit prime.[3] In total, this resulted in $4 \times 4 \times 2 \times 2 = 64$ combinations, each of which was presented five times per block (*i.e.,* $5 \times 64 = 320$ trials per block). On each trial, the left/right position of the cued location was randomly assigned, as well as the orientation of the digit and letter stimuli (*i.e.,* no stimulus could be tilted, one, or both). Both blocks began with random training trials which were discarded from all analyses (participants 1–3: 32 training trials per block; participants 4–29: 16 training trials per block.) Between the two blocks and every 64 trials, participants could take self-paced breaks.

### Stimulus contrast

Prior to the main experiment, visual contrasts of CF-suppressed digit/letter stimuli were determined for each participant individually to avoid regression artifacts arising from the post hoc selection of trials and/or participants based on residual stimulus awareness (*Rothkirch, Shanks & Hesselmann, 2022*; *Shanks, 2017*). Post hoc data selection is particularly problematic for studies using CFS, where suppression dynamics may vary dramatically between participants (*Yamashiro et al., 2014*), and residual stimulus visibility is common (*e.g.*, *Rothkirch & Hesselmann, 2018*).

To determine the highest contrast at which stimuli could not be seen anymore due to suppression by the CFS masks, stimulus contrasts were adjusted using a logarithmic function in a 1-up-1-down procedure (*Handschack et al., 2022*). Digits, letters, and cue stimuli were presented as in the main experiment; in each trial, participants were asked to report the subjective visibility of the attended stimulus using a binary response (seen, not seen). Central target stimuli were not presented. As described in the study by *Handschack et al. (2022)*, stimulus contrasts were varied by alpha blending with a uniformly grey image; the alpha value refers to the varying opacity of the grey image (0 = transparent; 1 = opaque). Starting off with stimuli at full contrast (alpha = 0), contrast decreased if participants reported "seen", and it increased if participants reported "not seen" (step multiplier = .7197). The staircase procedure terminated once participants performed a total of 25 trials. We repeated this sequence, if necessary, as some participants required more time to get used to the task. The digit/letter stimulus contrast in the main and control experiments was then set to the highest stimulus contrast that the participant consistently judged as invisible in the staircase procedure. The mean resulting alpha value was 0.72 (range: 0.23–0.96), the mean Michelson contrast was 0.21 (range: 0.02–0.65; $N = 25$).

### Control experiments

Conscious perception of CF-suppressed stimuli may increase when performing several trials on the same task (*Ludwig et al., 2013*). To rule out that prime digits were visible in CFS trials towards the end of the main experiment, we conducted the first control experiment. Stimuli were presented as in the main experiment (Fig. 1), with the exception that no targets appeared after the presentation of the digits and letters. Participants had to report whether the presented digit was smaller or larger than five. Next, the subjective visibility of the attended stimulus had to be rated. Responses were registered *via* button press, following the procedures of the main experiment. Each combination of prime digit, target digit, visibility level, and attended location was presented twice (*i.e.,* $2 \times 64 = 128$ trials). Stimulus orientation was randomly assigned on each trial. Trials were presented in random order, and participants could take a self-paced break after 64 trials.

In the second control experiment, we aimed to estimate the prime stimulus visibility in the absence of any conflicting visual information (*i.e.*, CFS masks), and therefore presented the digits and letters monocularly, at the individually determined contrast levels, but without the CFS masks. Stimuli and tasks were the same as in the first control experiment, with the exception that no CFS masks were presented. Each combination of prime digit,

target digit, and attended location was presented once (*i.e.,* $4 \times 4 \times 2 = 32$ trials; for participants 1–3 we used two repetitions per combination $= 64$ trials).

### Behavioural data analysis

Data pre-processing, descriptive and inferential statistics were performed using Matlab2019a (MathWorks Inc., USA), as well as R 4.1.1 (*R Core Team, 2021*) and RStudio 2021.09.0, Build 351 (*RStudio Team, 2021*). Data visualization was created with the R package ggplot2 (*Wickham, 2016*). Raw data in csv-format and R code are available at an online repository (https://doi.org/10.17605/OSF.IO/95AYU).

Data from training trials were discarded prior to any data analysis. Furthermore, behavioural data from three participants (#12, #15, #16) were discarded because all three participants did not follow the instructions, but performed the number comparison task on the prime stimulus; as a consequence, this resulted in approximately 50% incorrect trials in the main experiment. One further participant (#3) was excluded because they gave more "clear experience" PAS $= 4$ ratings than the other ratings in the CFS-masked trials of the main experiment (Fig. 2A, left panel, dotted line). This participant's data are only shown in the PAS distributions in Fig. 2AB, but were not used for any further analyses and inferential statistics (*i.e.,* total $N = 25$). Please note that there are no qualitative differences when this participant is included in the analyses. Analysis of RTs was restricted to trials with correct responses in the "smaller or larger than five" task. We used the interquartile range (IQR) method (*Tukey, 1977*) to define trials with RTs located 1.5 IQR outside the lower and upper quartiles as outliers (per participant, across all conditions). For frequentist null hypothesis significance testing (NHST), mean RT data were submitted to rm-ANOVA (factors: "prime-target congruency", "attention") using the afex package in R.

## Bayesian statistics

As an alternative to null hypothesis significance testing (NHST), we analyzed our data using Bayes factors (BFs). BFs describe the relative probability of data under competing positions (*e.g.,* a null model $H_0$ and one alternative model $H_1$). Specifically, the BF refers to the ratio of marginal likelihoods of different models under consideration, and quantify the change from prior to posterior model odds. The prior odds describe the beliefs about the models before observing the data. The BF thus describes how the evidence from the data should change beliefs (*Rouder et al., 2012*). The subscripts on BFs refer to the models being compared, with the first and second subscript referring to the model in the numerator and denominator, respectively. For example, a $BF_{10}$ of 4 indicates that the data are four times more likely under H1 than under H0. To calculate BFs, we used the generalTestBF function from the BayesFactor package (version 0.9.12-4.3) in R. Models, priors, and methods of computation are provided in *Rouder et al. (2012)*. GeneralTestBF uses Cauchy priors (scale parameter for standardized fixed effects: 0.5; random effects: 1; slopes: sqrt(2)/4). For the sake of conciseness, we refrained from calculating additional robustness analyses with different priors.

[4]Please note that the eye tracker's maximal sampling rate was 250 Hz. In combination with the mirror stereoscope, a sampling rate of 100 Hz turned out to yield the most robust tracking performance.

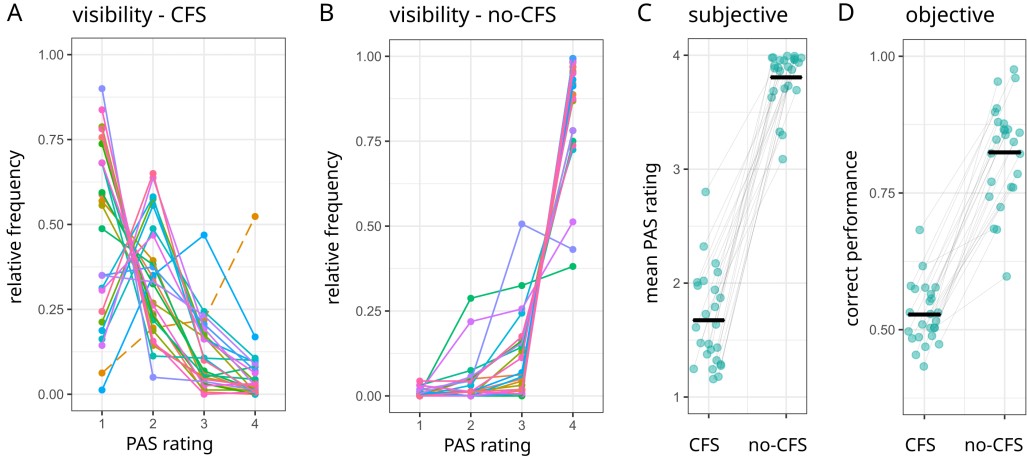

**Figure 2** **Visibility ratings and orientation discrimination performance in the main experiment.** (A) Distributions of PAS ratings for each participant (color-coded). Data from trials in which letters and digits were suppressed by CFS masks. Participant #3 was excluded from all further analyses, because CFS did not lower the visibility ratings as intended (broken line). (B) Data from trials in which stimuli were superimposed onto the CFS masks ("no-CFS" trials). (C) Subjective visibility. Mean PAS ratings (horizontal black lines) and individual data (green dots) for CFS and no-CFS trials. (D) Objective visibility. Mean correct performance (horizontal black lines) and individual data (green dots) in the orientation task, separately for CFS trials and no-CFS trials.

## Eye tracking

Eye tracking data were collected with a video-based eye tracker (Cambridge Research Systems, UK; spatial accuracy: 0.05°). Due to technical difficulties, eye tracking data were only recorded in 20 participants. The sampling rate was 100 Hz. In two participants, the sampling rate was 200 Hz, and in one participant it was 250 Hz; in these cases, the data were down-sampled to 100 Hz.[4]

In a first step, we eliminated all data points that exceeded the dimensions of the screen. Additionally, constant trends were removed by horizontal and vertical drift correction for each run. For noise reduction, we applied a low-pass filter with a sliding window of five data points. Participants were excluded from further analysis if available gaze data represented less than 5% valid data during prime presentation (Fig. S2). Coordinates of the fixation cross were calculated individually for each participant by the mean gaze coordinates during fixation screens only. Successful fixation was defined by detected gaze positions on the horizontal axis within 1.71° of visual angle from the fixation cross, corresponding to the distance of the largest stimulus from the fixation cross, while prime stimuli were presented. Within this period, fixation performance was computed as the percentage of successful fixation of the recorded gaze data.

Data from 18 participants were included in the eye tracking analysis (Fig. S2). On average, 98.9% ± 0.4 SEM of recorded data points during stimulus presentations were located within the defined fixation area. Figure S3 shows the spatial distribution of eye gaze during stimulus presentation intervals. Taken together, the eye tracking data indicate that participants showed successful fixation behaviour during the main experiment.

## RESULTS—MAIN EXPERIMENT

For the analysis of subjective visibility ratings, we computed the relative frequency of each PAS level (1–4), separately for each participant. Figure 2 (panels A and B) plots the resulting distributions per visibility condition and participant. As intended, trials in which letters and digits were suppressed by CFS masks were associated with lower PAS levels (Fig. 2A) than trials in which stimuli were superimposed onto the CFS masks (Fig. 2B). Overall, participants used the intermediate PAS levels (2,3) more often in CFS trials than in no-CFS trials. Figure 2C shows that the mean PAS rating in CFS trials was 1.68 ± 0.08 (mean ± SEM), while it was 3.79 ± 0.05 in no-CFS trials. Similarly, performance in the orientation discrimination task varied as a function of CFS condition: The mean orientation discrimination performance in CFS trials was 53% ± 1, while it was 81% ± 2 in no-CFS trials (Fig. 2D). Taken together, the data suggest that stimulus visibility was successfully reduced using CFS (*i.e.,* interocular suppression).

For the RT analysis ("compare target number to 5" task), only trials with correct responses to target numbers (97% ± 0.37) were included. We excluded all trials with RT outliers (based on the IQR method), which applied to 7% ± 1 of the correct trials.

In CFS trials, we expected that priming effects for unattended prime stimuli would be larger than priming effects for attended stimuli, as predicted by the "CFS-attenuation-by-inattention" hypothesis. However, we observed only small differences between RTs in incongruent and congruent trials, which were not affected by spatial attention (Fig. 3A, left panel). With primes at the unattended location, mean RT in congruent trials was 797 ms ± 9, and 804 ms ± 9 in incongruent trials (mean ± SEM). With primes at the attended location, mean RT in congruent trials was 795 ms ± 10, and 805 ms ± 6 in incongruent trials (Fig. 3B, left panel). Thus, the observed "CFS-attenuation-by-inattention" effect was 7–11 ms = −4 ms (95% CI [−26; 19]). A two-way rm-ANOVA indicated no main effect of prime-target congruency ($F(1, 24) = 2.79$, $p = .108$, $\eta_p^2 = .104$) or attention ($F(1, 24) < 1$), and no significant interaction between both factors ($F(1, 24) < 1$). Taken together, the results from our spatial cueing task do not support the "CFS-attenuation-by-inattention" hypothesis.

In no-CFS trials, we expected that prime-target congruency would facilitate performance in the "compare number to 5" task. With primes at the unattended location, mean RT in congruent trials was 849 ms ± 10, and 830 ms ± 7 in incongruent trials (Fig. 3A, right panel). With primes at the attended location, mean RT in congruent trials was 853 ms ± 9, and 837 ms ± 8 in incongruent trials (Fig. 3B, right panel). A two-way rm-ANOVA indicated a significant main effect of prime-target congruency ($F(1, 24) = 9.41$, $p = .005$, $\eta_p^2 = .282$), but not of attention ($F(1, 24) = 1.28$, $p = .268$, $\eta_p^2 = .051$), and no significant interaction between both factors ($F(1, 24) < 1$). Thus, to our surprise, we observed the inverse pattern (*i.e.,* negative priming effects), both for attended and unattended prime stimuli. When the congruency effects in CFS and no-CFS trials were directly compared in a three-way rm-ANOVA (factors: visibility, congruency, attention), the "visibility × congruency" interaction turned out to be significant ($F(1, 24) = 7.96$, $p = .009$, $\eta_p^2 = .249$), confirming that the congruency effect goes in the opposite direction in no-CFS trials.

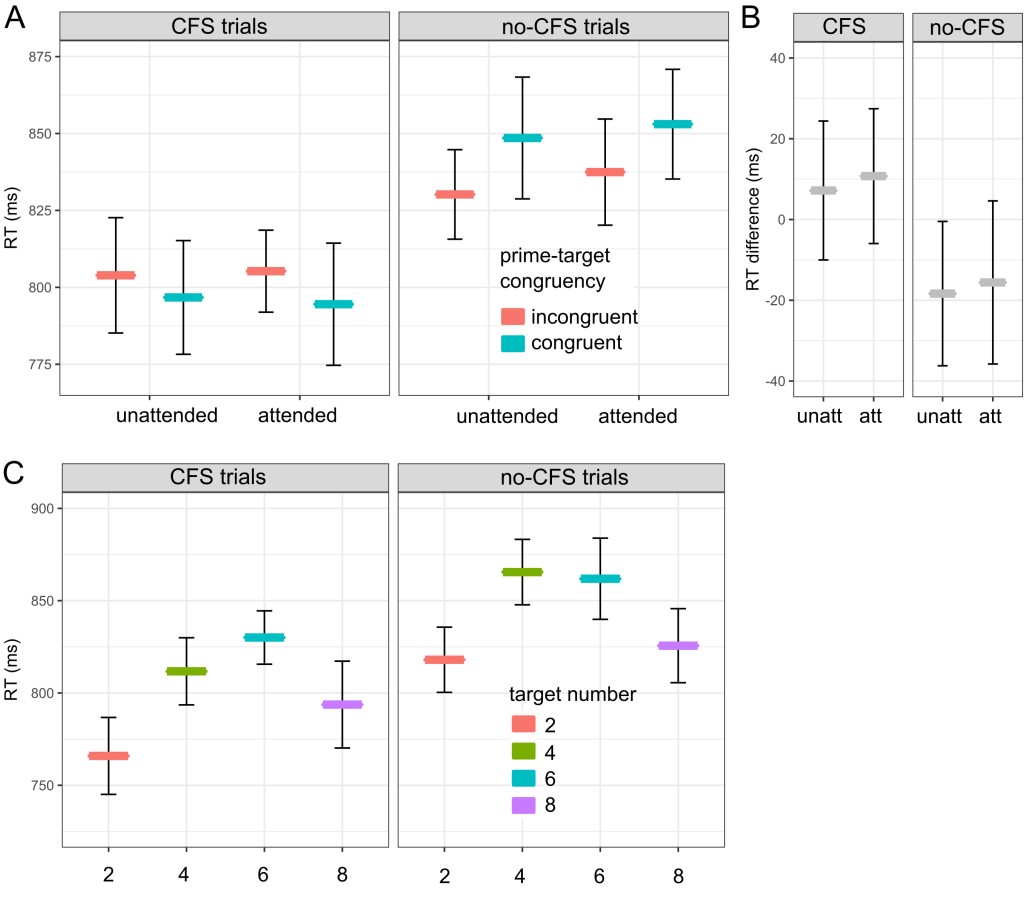

**Figure 3** **RT results.** Mean RTs ($N = 25$) from the "compare the target number to five" task in the main experiment. (A) RTs in CFS trials ("prime invisible", left panel) and no-CFS trials ("prime visible", right panel), separately for prime-target incongruent trials (red) and congruent trials (blue) in the unattended and attended conditions. (B) RT difference: RT for incongruent trials minus RT for congruent trials, separately for unattended (unatt) and attended (att) conditions. (C) RTs in CFS trials ("prime invisible", left panel) and no-CFS trials ("prime visible", right panel), separately for each target number (2, 4, 6, 8). Error bars indicate 95% confidence intervals calculated for within-subject data using the summarySE within function from the Rmisc package (version 1.5.1).

As we did not expect the inverse priming effect in no-CFS trials, we decided to investigate its origins in a number of exploratory analyses. It has been shown that some published numerical priming effects may be due to confounds in the experimental design, *e.g.*, when target-related differences in RTs are not taken into account (*Hesselmann et al., 2015*; *Hesselmann & Knops, 2014*). Figure 3C indicates that our data contain a target-specific distance effect, *i.e.,* target numbers closer to 5 were associated with longer RTs than target numbers with a larger numerical distance from 5, both in CFS trials and no-CFS-trials. (Please remember that participants' task was to compare the target number to five.) However, this distance effect cannot have acted as a confound in our experimental design, because all target numbers (2, 4, 6, 8) were equally often paired with congruent and incongruent numerical primes. Figure 3C also shows that trials in no-CFS conditions were

associated with longer RTs than trials in CFS conditions. Differences in prime visibility, however, can be ruled out as a confound in our experimental design because all prime numbers (1, 3, 7, 9) were equally often paired with congruent and incongruent numerical targets.

In our previous study investigating the "CFS-attenuation-by-inattention" hypothesis by means of numerical priming (*Benthien & Hesselmann, 2021*), some participants told us during the debriefing that they had experienced an unexpected response conflict in the main experiment, in particular when the primes were visible in the no-CFS trials. The same sets of primes and targets were used in this earlier study. Participants were instructed to perform the same number comparison task, but the numerical primes were presented either at one location (focused attention, location certain), or at one out of two possible locations (diverted attention, location uncertain). Participants reported that they had compared the target to the prime stimulus as well, which resulted in a response conflict for specific prime-target pairs: Whenever the target was larger than the prime, but smaller than five (*e.g.*, prime: 1; target: 2), or when the target was smaller than the prime, but larger than five (*e.g.*, prime; 9; target: 8), the response was more effortful because the target-prime relation (smaller, larger) and the number comparison task ("compare target to five"; smaller, larger) were in conflict. In other words, whenever the prime-target sequence was getting larger (or smaller), the opposite response in the number comparison task (*i.e.,* "smaller than five", or "larger than five") was experienced as more effortful. Following the approach in our previous study, we aimed to elucidate the potential response conflict with the help of Bayesian linear mixed effects models (LMMs) applied to trial-by-trial RTs.

As in our previous study (*Benthien & Hesselmann, 2021*), we first assigned all 16 prime-target pairs to a "conflict" condition (six prime-target pairs: 1–2; 1–4; 3–4; 7–6; 9–6; 9–8), and a "no-conflict" condition (ten prime-target pairs: 1–6; 1–8; 3–2; 3–6; 3–8; 7–2; 7–4; 7–8; 9–2; 9–4). Next, within the "conflict" condition, we further differentiated between prime-target pairs with either small or large numerical distances (small: 1–2, 3–4, 7–6, 9–8; large: 1–4, 9–6). Finally, in a new variable, we coded the "no conflict" condition as 0, and the small and large "conflict" conditions as 1 and 2, respectively. Mean RTs were 831 ms $\pm$ 5 in "no conflict" trials, and 854 ms $\pm$ 7 and 878 ms $\pm$ 7 in the two "conflict" conditions. The RT difference between "no conflict" and "large conflict" trials was 47 ms (95% CI [30; 64]).

We calculated Bayesian linear mixed effects models (LMMs), including participants and target stimuli as random intercepts (*Benthien & Hesselmann, 2021*; *Moors & Hesselmann, 2018*). Prime-target congruency and response conflict were included as fixed factors. From all calculated models, we extracted the model with the highest BF compared to an empty model (*i.e.,* an intercept-only model) and considered this to be the best fitting model. We then recalculated all BFs such that they were compared to this model (*Heyman & Moors, 2014*). In Table 1, as described in *Benthien & Hesselmann (2021)*, this yields a summary of the best fitting model (BF = 1) for visible (no-CFS) trials, and how much more likely the data are under this model than under all other calculated models. Specifically, BFs > 1 indicate how much more likely the data are under the best fitting model compared to the model under consideration.

**Table 1** Bayesian statistics. Bayes factors (BFs) for the first exploratory analysis ($N = 25$). Linear mixed effects models for RT data in no-CFS trials included participants ("subj") and target stimuli ("target") as random intercepts, and prime-target congruency ("cong") and response-conflict ("conflict") as fixed factors. BFs were calculated with the best fitting model in the numerator (the model for which BF = 1, top row). BFs > 1 indicate how much more the data are consistent with the best fitting model compared to the model under consideration.

| No-CFS trials (visible) | BF | Error |
|---|---|---|
| cong + subj + target | 1.000000 | 0.0000000 |
| conflict + subj + target | 1.948036 | 0.0185061 |
| subj + target | 2.286223 | 0.0151891 |
| cong + conflict + subj + target | 41.087761 | 0.0377258 |
| All other models | >100 | NA |

Table 1 shows that the best fitting model in no-CFS (visible) trials included participant and target stimulus as random intercepts, and prime-target congruency as fixed effect. This result is in good agreement with the rm-ANOVA, but the best fitting model was only weakly (BF = 1.9) favored over a model that included response conflict instead of congruency as fixed effect, thus suggesting a contribution of both factors. The next model, however, including only participant and target stimulus, reached a similar BF of 2.3, suggesting only a weak influence of response conflict. Finally, the observed data were much more likely (41.1 times) under the best fitting model than under a model including all random intercepts and fixed effects. In CFS trials, the best fitting model included only participant and target stimulus as random intercepts. Taken together, the results from our first exploratory analysis seem to be in line with our previous study (*Benthien & Hesselmann, 2021*), but do not constitute compelling evidence of response conflict in no-CFS trials.

In our second exploratory analysis, we aimed to further disentangle the contribution of prime-target congruency and response conflict on RTs. This proved to be difficult, as both predictors were correlated: all incongruent trials were also trials without potential response conflict (prime-target pairs: 1–6, 1–8, 3–6, 3–8, 7–2, 7–4, 9–2, 9–4). Among congruent prime-target pairs, six were associated with a potential response conflict (1–2, 1–4, 3–4, 7–6, 9–6, 9–8), and two were not (3–2, 7–8). We therefore decided to limit the BF-LMM analysis to four congruent prime-target pairs, two with a potential response conflict (1–2, 9–8) and two without (3–2, 7–8), thus keeping the targets constant across conflict conditions. To increase statistical power, we included the RT data from our earlier study (*Benthien & Hesselmann, 2021*), resulting in total $N = 51$, and total number of observations = 3459. Mean RTs were 651 ms ± 7 in "no conflict" trials, and 663 ms ± 7 in "conflict" trials. The RT difference was 12 ms (95% CI [−7; 31]). Table 2 shows that the best fitting model in no-CFS (visible) trials included only participant as random intercept. The observed data were more likely (11.3 times) under the best fitting model than under a model also including response conflict as fixed effect, thus speaking against the hypothesis of a response conflict, at least for this selection of trials.

Finally, we conducted one further analysis to explore the effects of the dual-task structure of the experiment. After the primary task (speeded response to the target), participants had to perform a secondary task: in one block, to judge whether the attended stimulus

**Table 2  Bayesian statistics.** Bayes factors (BFs) for the second exploratory analysis ($N = 51$). Linear mixed effects models for RT data in no-CFS trials included participants ("subj") and target stimuli ("target") as random intercepts, and response-conflict ("conflict") as fixed factor. BFs were calculated with the best fitting model in the numerator (the model for which BF = 1, top row). BFs > 1 indicate how much more the data are consistent with the best fitting model compared to the model under consideration.

| No-CFS trials (visible) | BF | Error |
|---|---|---|
| subj | 1.000000 | 0.0000000 |
| subj + target | 3.095224 | 0.0101445 |
| conflict + subj | 11.323016 | 0.0277006 |
| conflict + subj + target | 36.143763 | 0.0350267 |
| All other models | >100 | NA |

was tilted or not, and in another block, to rate the visibility of the attended stimulus on a four-point awareness scale (PAS). One could predict that the potentially higher processing requirements in the PAS task will negatively affect the response speed in the primary task. We therefore analyzed the effects of prime-target congruency (two levels: congruent, incongruent), and second task (two levels: tilt, PAS), separately for CFS and no-CFS trials. The results are shown in Fig. S4. To our surprise, RTs in "tilt" trials turned out to be approx. 100 ms longer than in "PAS" trials, and this RT increase was limited to the "prime visible" (no-CFS) trials. In fact, the observed RT differences between "prime invisible" and "prime visible" trials (shown in Fig. 3) appear to be mainly driven by the long RTs in the "prime visible" trials with the tilt task as second task. In terms of priming effects, visual inspection of Fig. S4 reveals that the effect of prime-target congruency was larger for "tilt" trials than for "PAS" trials. In visible no-CFS trials, this was confirmed by a significant "congruency × task" interaction ($F(1, 24) = 5.82$, $p = .024$, $\eta_p^2 = .195$), but not in CFS trials ($F(1, 24) = 2.95$, $p = .099$, $\eta_p^2 = .109$). One plausible explanation could be that only the objective tilt task forces participants to follow the cue, as opposed to the subjective PAS task.

## RESULTS—CONTROL EXPERIMENTS

In the first control experiment, we checked whether the CF-suppressed stimuli were more visible after the main experiment. Participants first had to report whether the presented digit was smaller or larger than five. In CFS trials, the mean performance was 62% ± 3 SEM, suggesting that the CF-suppression of prime identity was not as strong as the suppression of tilt (53% accuracy in the main experiment), while it was 95% ± 2 SEM in no-CFS trials. Whether the digits appeared at the attended or unattended location had only a negligible influence on the performance levels (mean difference <1%). Next, the subjective visibility of the attended stimulus had to be rated. In CFS trials, the mean PAS rating was 1.53 ± 0.08 SEM. In no-CFS trials, where the stimuli were superimposed on the CFS masks, the mean PAS rating was 3.86 ± 0.03 SEM. In sum, the results indicate that the depth of CF-suppression was not attenuated over the course of the main experiment.

In the second control experiment, we tested whether the low-contrast digits and letters were fully visible when no CFS masks were presented. Participants first had to report

whether the presented digit was smaller or larger than five. The mean performance was 97% ± 1 SEM. Whether the digits appeared at the attended or unattended location had only a negligible influence on the performance levels (mean difference <1%). Next, the subjective visibility of the attended stimulus had to be rated. The mean PAS rating was 3.80 ± 0.07 SEM. As expected, the results thus show that without the CFS masks, the low-contrast stimuli were fully visible.

[5] As we have pointed out elsewhere (*Handschack et al., 2022*), experiments on unconscious processing often seek to maximize both the bottom-up signal intensity and the invisibility of stimuli. This has proven to be difficult. Care must be taken that the assumption of a "sweet spot" (where both parameters are maximal) does not lead to a situation where the absence of evidence for unconscious processing is exclusively attributed to low signal intensity (*e.g.*, low stimulus contrast).

## DISCUSSION

This study aimed to test the "CFS-attenuation-by-inattention" hypothesis (*Eo et al., 2016*) by investigating numerical priming under CFS in a spatial cueing paradigm. According to this hypothesis, visuo-spatial attention—when allocated to a specific location *via* cueing—increases interocular suppression, and hence a stimulus at that location does not undergo high-level processing; conversely, when attention is diverted away from this location, this leads to a relaxation of interocular suppression, and hence an increase in high-level semantic processing. Our RT data provide no evidence for unconscious numerical priming under CFS, irrespective of where the primes were presented, and thus do not support the "CFS-attenuation-by-inattention" hypothesis. In no-CFS trials with visible primes, RTs showed a negative (or, inverse) priming effect, with congruent primes being associated with longer RTs than incongruent primes. In the following, we will discuss these findings and further exploratory results.

While the ERP study by *Eo et al. (2016)* showed an N400 effect for word primes presented at an uncued location, and thus evidence for semantic processing, we failed to find priming effects for numerical primes presented at an uncued location. How can these different study outcomes be explained? First, one could argue that, although both studies involved CFS and a spatial cueing paradigm, critical differences remain between the setups that make the results from both studies hard to compare, such as the exact spatio-temporal parameters of the CFS masks. *Eo et al. (2016)* used masks that were very similar to their word stimuli, whereas we used Mondrian-like CFS masks to suppress Arabic number symbols. In light of extensive previous work showing that shared characteristics of CFS masks and stimuli can increase the level of interocular suppression (*Hong & Blake, 2009*; *Zadbood, Lee & Blake, 2011*; *Yang & Blake, 2012*), it is conceivable that the suppression level was indeed stronger in the original ERP study than in our study. Second, one could argue that in our study the suppression of the prime stimuli by the CFS masks was too strong, or that the prime contrasts were set too low, and this is why participants could not process the semantic/numerical information of the prime stimuli. Although our data show that prime stimuli were fully visible without the CFS masks, and that primes were in fact bordering on being too visible in CFS trials (with objective performances at 53% and 62%), stimulus contrast and thus stimulus strength remains to be an important factor that might explain different outcomes in studies on unconscious visual processing. Overall, it has been proven difficult to compare and integrate CFS results due to such differences in study design which might influence CF-suppression depth (*Ludwig & Hesselmann, 2015*).[5] Follow-up studies on unconscious processing under CFS will therefore benefit from more standardized

procedures. Finally, it may be the case that the N400 effect observed by *Eo et al. (2016)* does not "translate" into observable behavioural effects, such as semantic priming. On the other hand, the results from this study, as well as from our previous priming study (*Benthien & Hesselmann, 2021*) and fMRI-MVPA study (*Handschack et al., 2022*), are in good agreement with the emerging view that the representation of CF-suppressed stimuli is fractionated, and limited to their basic, elemental features (*Moors et al., 2019*; *Moors et al., 2017*). Results from another study investigating distance-dependent numerical priming point into the same direction (*Hesselmann et al., 2015*).

In no-CFS trials with visible primes, our results show a significant priming effect, albeit in the direction opposite to what was expected (*i.e.*, a negative or inverse priming effect: RT congruent > RT incongruent). To the best of our knowledge, inverse priming effects are observed less frequently than positive priming effects. In response priming studies (*e.g.*, using metacontrast masking), positive priming effects typically occur for prime-target SOAs of up to 100 ms (*Neumann & Klotz, 1994*), but for longer SOAs response-priming effects may reverse (*Lingnau & Vorberg, 2005*). In a masked face repetition priming paradigm using CFS (*Barbot & Kouider, 2012*), inverse priming effects were observed when primes were presented for a prolonged duration of 1,000 ms. The authors refer to this effect as "nonconscious overstimulation cost", based on a neural habituation priming model that explains the change from positive to negative priming with increasing prime duration (*Huber & O'Reilly, 2003*). In a recent study investigating priming of natural scene categorization during CFS (*Baumann & Valuch, 2022*), positive prime-target congruency effects were observed for the shortest RTs, confirming the presence of response priming, whereas inverted congruency effects were observed for the longest RTs, suggesting that response inhibition processes were also at work.

Based on our previous numerical priming study (*Benthien & Hesselmann, 2021*), we explored an alternative hypothesis to explain the negative priming effect. In our previous study, participants reported that they had compared the target to the prime stimulus as well, which resulted in the experience of a response conflict for specific prime-target pairs ("conflict" hypothesis). Data from our current study does not support this hypothesis. In our research design, however, all incongruent trials were also trials without potential response conflict, so that the results from our exploratory analyses are not conclusive. To better understand the observed inverse priming effect, follow-up studies should therefore aim to disentangle the contribution of SOA, prime-target congruency and response conflict on RTs.

We observed that RTs in no-CFS trials with visible primes were approx. 50 ms longer than RTs in CFS trials with invisible primes. (Please note that this RT difference appears to have been mainly driven by long RTs in no-CFS trials where the tilt discrimination task was the second task; see Fig. S4). This RT difference may be due to dual-task costs triggered by the visible prime stimulus and the prime-related task (*e.g.*, the PAS rating). Overall, RTs in our study (800–850 ms) turned out to be considerably longer than RTs in similar semantic priming studies, in which average RTs around 500 ms were observed (*Benthien & Hesselmann, 2021*; *Dehaene et al., 1998*). We speculate that this difference is related to the fact that in our study participants were confronted with two different prime-related tasks

(visibility rating; orientation discrimination), albeit one per block. This additional cognitive load may have further increased the dual-task costs, triggered by prime onset. Together with the relatively long prime-target SOA in our study (200 ms), there is the question whether positive priming effects are to be expected at all, or whether the results are more in the range of inverse priming effects, *i.e.*, negative compatibility effects (*Baumann & Valuch, 2022*; *Lingnau & Vorberg, 2005*).

Another remaining question is whether the priming task used in our study is semantic at all. One could argue that any priming effects observed in this task can be taken as evidence for an elaborate processing of unconsciously presented numerical information (*Dehaene et al., 1998*). However, as the task used only two digits for each of the two response categories, one could instead argue that participants quickly learned the fixed S-R associations necessary for the task, thus rendering it a response priming task (*Henson et al., 2014*). For example, *Kunde, Kiesel & Hoffmann (2003)* proposed that the priming effects observed in a similar task were due to a "match with pre-specified cognitive action-trigger conditions" (p. 223). The differentiation between semantic and response priming may have direct implications for the magnitude of the priming effect in our study. In order to decrease the visibility of the prime, we degraded the prime signal by adjusting the prime's visual contrast individually for each participant. In response priming, prime contrast determines the size of the priming effect: the larger the prime contrast, the larger the priming effect (*Schmidt, Haberkamp & Schmidt, 2011*). If we assume that our task is a response priming task, the reduced prime contrasts could then explain why we did not observe any priming effects in the "invisible prime" conditions. Please note, however, that in the "visible prime" conditions the prime stimulus was presented at full contrast; therefore, the relationship between prime contrast and the size of the priming effect cannot explain the negative (or, inverse) priming effect we observed in no-CFS trials with visible primes.

One limitation shared by our study, a previous spatial cueing experiment co-authored by one of us (*Benthien & Hesselmann, 2021*), and the original ERP study by *Eo et al. (2016)* is the lack of an independent attentional manipulation check. Without such a manipulation check, how can we be sure that participants actually shifted their attention based on the central cue? In our previous fMRI-MVPA study, we could successfully decode the attended side from striate cortex, which suggests that participants indeed directed their attention as instructed (*Handschack et al., 2022*). A previous behavioral CFS study used visible stimuli during each trial to verify that spatial attention was indeed being deployed, so that any differences in the processing of CF-suppressed stimuli could be attributed to the effect of visuospatial attention (*Bahrami et al., 2008*). If feasible, future studies aimed at investigating the "CFS-attenuation-by-inattention" hypothesis should employ a similar experimental design.

## CONCLUSIONS

To put it simply, our data provide no support for the "CFS-attenuation-by-inattention" hypothesis and suggest that there might be "not much to see" under CFS, or even no number processing outside awareness per se (*Zerweck et al., 2021*). It is conceivable that

the large heterogeneity between published CFS findings is due to other factors and choices, *e.g.*, the exclusion of participants and trials due to residual stimulus awareness, which may lead to regression-to-the-mean artifacts (*Rothkirch, Shanks & Hesselmann, 2022*; *Shanks, 2017*). From a broader perspective, we believe that future CFS and masked priming studies should consider the mapping of suppression methods within a functional hierarchy of unconscious processing (*Breitmeyer & Hesselmann, 2019*; *Breitmeyer, 2015*), as this will help to constrain the generation of new hypotheses.

## ACKNOWLEDGEMENTS

The authors would like to thank Peter Heinze for his invaluable assistance with participant recruitment and data collection, and Wanyi Lyu for her help with the eye-tracking data analysis. We would also like to thank the reviewers for their helpful and constructive comments.

### Funding
This work was supported by the German Research Foundation (DFG grant HE 6244/1-2). The funders had no role in study design, data collection and analysis, decision to publish, or preparation of the manuscript.

### Grant Disclosures
The following grant information was disclosed by the authors:
German Research Foundation: DFG grant HE 6244/1-2.

### Competing Interests
Guido Hesselmann is an Academic Editor for PeerJ.

### Author Contributions
- Juliane Handschack conceived and designed the experiments, performed the experiments, analyzed the data, prepared figures and/or tables, authored or reviewed drafts of the article, and approved the final draft.
- Marcus Rothkirch conceived and designed the experiments, authored or reviewed drafts of the article, and approved the final draft.
- Philipp Sterzer conceived and designed the experiments, authored or reviewed drafts of the article, and approved the final draft.
- Guido Hesselmann conceived and designed the experiments, analyzed the data, prepared figures and/or tables, authored or reviewed drafts of the article, and approved the final draft.

### Human Ethics
The following information was supplied relating to ethical approvals (*i.e.*, approving body and any reference numbers):

This study was approved by the Ethical Committee of the German Association of Psychology (Deutsche Gesellschaft für Psychologie, DGPs; ethical application ref: GA_Hesselmann-092010-rev).

### Data Availability

The data and code are available at OSF: Hesselmann, Guido. 2022. "CFS Number Priming & Spatial Cueing". OSF. November 9. doi: https://www.doi.org/10.17605/OSF.IO/95AYU.

### Supplemental Information

Supplemental information for this article can be found online at http://dx.doi.org/10.7717/peerj.14607#supplemental-information.

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
