# Peer review of "No effect of attentional modulation by spatial cueing in a masked numerical priming paradigm using continuous flash suppression (CFS)"

_PeerJ, doi:10.7717/peerj.14607_

## Round 0.1 · original submission · Major Revisions

As you can see, I have now received the comments from three experts (one Reviewer preferred to remain anonymous, the other two are Sascha Meyen and Thomas Schmidt). The three expert reviews offer detailed and insightful feedback regarding your manuscript, for which I am incredibly grateful. As you can see from their feedback, all reviewers have offered a generally positive assessment of your manuscript and felt that it is well-positioned to make a substantial and valuable contribution to the literature. I agree with this positive evaluation.

All expert reviews are precise and clear and have offered several concrete suggestions for improving your manuscript. I will, therefore, not reiterate all their points here. But I ask you to consider each of them; in particular, both Reviewers 2 and 3 asked for additional analyses and Reviewer 1 also suggested some theoretical improvements. With this in mind, I would also like to ask you to consider all the following points when revising the manuscript to make it eligible for publication in PeerJ.

·

Basic reporting

all requirements met

Experimental design

all requirements met

Validity of the findings

see review below

Additional comments

Review of Ms. #76930, "The scope and limits of numerical priming under interocular suppression – No effect of attentional modulation by spatial cueing", by Juliane Handschack, Marcus Rothkirch, Philipp Sterzer, and Guido Hesselmann

Reviewer: Thomas Schmidt

The authors present an experiment (with two control experiments) designed to test a counterintuitive hypothesis in Continuous Flash Suppression (CFS): namely, that withdrawing attention from a prime stimulus can increase the priming effect because it lowers the effect of the CFS mask. To test this, they combine a priming task on digit classification (prime, target </> 5) with a cueing paradigm. Prime visibility is assessed in a secondary task on the same trial using a PAS-type scale. The authors find only small and insignificant priming effects when CFS is employed, but significant reverse priming when no CFS is employed. Priming effects are not modulated by the attentional manipulation. It is concluded that the CFS attenuation hypothesis is not valid.

I find the paper well-written, transparent and solid, yet have a few critical remarks.

MAJOR

1) One major question is whether the employed priming task is semantic at all. It uses only two digits for each of the two response categories, and participants would quickly learn the fixed S-R associations necessary for the task. That would render it standard response priming that required no actual numerical comparison with the number 5. (the same criticism applies to Dehaene's famous paper, of course). However, as a test of the attenuation hypothesis, this wouldn't matter much.

2) Some methodological aspects of the experiment make it more difficult to find positive priming effects. Prime contrasts were equated for each observer in a staircase procedure. This led to large variations in Michelson contrast, ranging from 0.02 – 0.65. As prime contrast should directly determine the size of the priming effect, one would expect large differences in priming effects between observers. Generally, the larger the prime contrast, the larger the priming effect, and the larger the target contrast, the shorter the reaction time. (Was it only the prime contrast that was adjusted this way or the target contrast, too?).

3) Responses to the target are surprisingly slow, probably because of the dual-task structure of the experiment. After the primary task (speeded response to the target), participants had to perform a secondary task: in one block, to judge whether the attended stimulus was tilted or not (which cleverly forces participants to process the attentional cue); in another block, to rate the visibility of the prime on a four-point scale. This latter task was quite cumbersome because it required selection via arrow keys, so that the processing requirements here could affect the speed of the primary task. In the results section, response time effects are not shown separately for these different secondary tasks, and I wondered whether the PAS trials were even slower than the tilt trials and whether there were differences in priming (e.g., because only the tilt secondary task forces participants to follow the cue). In any way, the overall dual-task structure leads to very slow responses to the target. Together with the long prime-target SOA (200 ms), there is a question whether positive priming effects are to be expected at all, or whether the results are more in the range of the Negative Compatibility Effect (reversed priming as found here in the no-CFS trials).

MINOR

4) l.31: "not all invisible stimuli are equally invisible". As much as I enjoy the paradox, I think the meaning of this sentence is not very clear, especially for an abstract.

5) l.37: "attention diverted away from the suppressed stimulus". It would be clearer and less perplexing to write something like "CFS attenuated by the removal of attention".

6) Footnote 1: The number of trials per condition is at least as important as the number of subjects.

7) l. 295: There are many published guidelines now for reporting Bayes Factors (e.g., Kruschke, 2021). Following them completely would be cumbersome, but some essentials are necessary. It is indispensable to clearly state what prior distributions were used, what parameters they had, and whether they are intended to be conservative or lenient with respect to the null and the alternative hypothesis. It is never enough to report that "defaults" were used, because those will differ between software packages and over time (for instance, JASP changed the dispersion parameter of its Cauchy prior from 1 to 0.717 to make it more sensitive, i.e., lenient in favor of the alternative hypothesis). Because the prior can completely determine the outcome of the test, it is necessary to state that it was chosen before knowing the data. Some textbooks additionally recommend robustness analysis (testing the limits when trying out different priors).

8) l. 347: Sounds as if the tilt task was intended to be a measure of prime visibility. But tilt is not the stimulus variable that generates the priming effect, so measuring tilt discrimination does not match the priming task. Isn't the tilt task only there to make sure that participants have to use the attentional cue? The later analysis of the prime discrimination task (control experiment 1) says that participants were 62 % correct on CFS trials, indicating that masking of prime identity was not as strong as masking of tilt.

9) l. 446: The authors write that "the model was 41.1 times less likely than the best-fitting model", but Bayes Factors can never be interpreted in terms of probabilities of models. All they give is a likelihood ratio: the odds of the DATA under one model relative to the odds of the data under a comparison model. To arrive at the posterior odds of the model itself (odds, not likelihood!), the Bayes Factor must be multiplied with the a priori odds of the model ("posterior odds = Bayes Factor x prior odds"). This is extremely important because the prior odds of the hypothesis can completely change the posterior odds even for a large BF. But in the present application of BFs with nested regression models, it would be particularly difficult to assign an a-priori probability to any one of those models to begin with.

·

Basic reporting

The manuscript is written in a clear and understandable way. The introduction relates to the relevant literature. Although minor improvements can be made, the structure of the manuscript is up to standards. Figures for the discussed effects are well prepared but additional visualizations can be generated to help understand the underlying data. The raw data is supplied by the authors but an update of the available material is required regarding a missing data file used in the analysis scripts.

Experimental design

The research question is well defined. With a liberal interpretation, one can consider this research as falling under Biological Sciences in terms of investigating the human visual system but it would certainly be an outlier in PeerJ regarding its scope. It appears to me that a replication of their experiment can be straightforwardly done based on the description of their methods. Regarding the analysis, I have some concerns that may be ruled out if the authors provide additional data and analyses. These concerns primarily focus on reporting measurement errors for the investigated effects instead of relying on ANOVA F values or Bayes Factors.

Validity of the findings

The motivation for the study's replication is thoroughly elaborated on and is clearly relevant to the current literature on unconscious processing in CFS. The data has been provided. The statistical analysis requires a bit of work and conclusions will depend on the improved analysis. For now, the link between interpretation and the original research question is not as strong as it could and should be.

Additional comments

The study conceptually replicates Eo et al. (2016) to investigate the hypothesis of whether continuous flash suppression is attenuated (priming effects enlarged) when attention is directed away from an invisible stimulus. No effects were found in the critical invisible condition: neither priming nor the targeted attenuation effect.

The study is written in a plain and understandable way. The experiment is well motivated. The failure to replicate is relevant for the field because it is (a) done with care for methodological details and (b) points to the heterogeneity of the established literature. Within the ongoing debate about unconscious processing, the study is a small yet relevant piece of evidence. Unfortunately, the study is limited in its interpretability regarding the targeted "CFS-attenuation-by-inattention" effect because no priming effect is found. Without a priming effect inferences about its attenuation is problematic. The authors need to discuss this problem more openly. Additional, some minor corrections are necessary.

MAJOR ISSUES

1. DISCUSS THE MISSING PRIMING EFFECT IN CFS TRIALS

The study did not find a prime-target congruency effect in CFS trials. This failure to replicate should be discussed more prominently:

(a) First, the measurement errors of the priming effects need to be presented and discussed. The authors already present mean RTs in Figure 3 indicating priming effects of 7 and 10 ms in the CFS condition. They also present results of a repeated measures ANOVA (line 361). The partial eta-squared appears quite large but the effect is nevertheless non-significant. I would like to be able to judge how precisely these effects were measured. Please add standard errors or confidence intervals. Please also highlight the power analysis by moving it from the footnotes to the main text and elaborate on the power of the basic priming effect.

In this step, it must be made clear whether there is evidence for the absence of an effect or whether sizeable priming effects cannot be excluded based on the data: What size of priming effects can be excluded based on the data? I would also like to see standard errors for no-CFS trials. Perhaps this information can be added to Figures 3A and B by adding a middle column for RT differences (incongruent minus congruent) in unattended and attended conditions, respectively.

(b) Second, assuming the authors present evidence for the absence of a priming effect, this should be more prominently discussed. I assume Eo et al. would claim that the "CFS-attenuation-by-inattention" effect is only to be expected when there is a substantial priming effect to begin with. Thus, the current study has limited relevance regarding the "CFS-attenuation-by-inattention". I believe it is appropriate to mention that early in the discussion.

(c) Nevertheless, failure to replicate the basic priming effect is relevant in its own right. Thus, third, the authors should focus their discussion on the elusiveness of priming effects for stimuli rendered invisible by CFS. This should accompany their point regarding the heterogeneity of results in the field (lines 576–577; please elaborate on this interpretation and put it earlier in the discussion). Whether stimuli were simply not visible enough to produce any effect (discussed in lines 505–531) should be augmented by the results. I believe the authors can argue that their stimuli were bordering on being too visible: Objective discrimination performance already deviates from chance level in the discrimination task (53% +1%, line 348) and Control Experiment 1 (62% +-3, line 476). Ideally, the authors include their subjective PAS measures here as well. Arguing that other studies achieved lower PAS scores could be considered evidence against the notion that stimulus strength was simply too low for even unconscious effects. Emphasizing the issue of the missing priming effect would improve the clarity of the discussion section.  
Please also refer to the interpretation made in the last sentence of the abstract about the suppression of _semantic_ processing under CFS in the main text again. With that in mind, the title is too broad for this result. It promises a comprehensive understanding of the "scopes and limits" of priming in CFS but only presents one failure to replicate—the study should be labeled as such and not more.
2. EFFECTIVE DESCRIPTIONS

While the writing style is very clear, in some parts, it is a bit inefficient.

(a) When Control Experiments 1 and 2 are described (245–268), four sentences are repeated word-for-word ("Stimuli were presented as in the main experiment [...] registered via button press, following the procedures of the main experiment."). Since the authors describe the two control experiments so closely together, I recommend describing them jointly to save space and facilitate understanding—I have been comparing word for word in the search for the difference between the two control experiments until I realized that the crucial difference pertains to the visibility condition. These two control experiments may also be shorted in the results section because their data is not leveraged much.

(b) Second, information about eye-tracking is dispersed. I understand that the authors want to separate the hardware information (lines 270–274), the analysis methods (312–322), and the results (333–339). I recommend collecting this information. For example, this could be done by merging the hardware information and the analysis method. Because the results are mostly supplementary and do not change interpretations, I can also imagine putting all information in the methods section. To avoid misunderstanding: I am happy that the authors report eye-tracking. But it is not worth it for the reader to keep track of it over three subsections.

(c) Third, regarding Bayesian statistics, there is some information about the analysis method in lines 295–310 and some in lines 420–431. They should be collected in the methods section. But I would ask the authors to keep the helpful reminders about interpretations when results are discussed. Moreover, I would ask the authors to elaborate on the priors. Sometimes, the alternative hypothesis prior can deteriorate into a straw man if it is catered towards very large effects while, in priming research, one would only expect small effects (see arguments Dienes, 2021). Because this information is currently missing, I am suspicious about the Bayesian evidence against the conflict effect. Therefore, I would like to see more exploratory analyses on the mean RTs and standard errors. I took a quick (potentially error-prone) look at the data myself and estimated the conflict effect to be 31 ms in the main data set. This seems relatively large and to be in the same range as the priming effect in the no-CFS trials. Whether the conflict factor is indeed negligible needs to be discussed more thoroughly. One way to present this data better could be highest density intervals (Kruschke, 2013) if the authors want to stay within the Bayesian framework here. But confidence intervals for the difference in the mean RTs would be sufficient for me.

Dienes, Z. (2021). How to use and report Bayesian hypothesis tests. Psychology of Consciousness: Theory, Research, and Practice, 8(1), 9, https://psycnet.apa.org/doi/10.1037/cns0000258.

Kruschke, J. K. (2013). Bayesian estimation supersedes the t test. Journal of Experimental Psychology: General, 142(2), 573. https://doi.org/10.1037/a0029146

MINOR ISSUES

3. The prime stimuli are described as being numbers 1, 3, 5, 7 (line 212 but also repeatedly later). Towards the end the 5 changes to a 9 and it is 1, 3, 7, 9. I assume 1, 3, 5, 7 is a typo.

4. It took me a moment to understand that "CFS-attenuation" refers to attenuation of the suppression (better processing of the stimulus) rather than suppression of processing the stimulus. I recommend the authors look for a way to make this terminology clear from the start. For example, the abstract (line 37-39) reads "CFS is facilitated whenever attention is diverted" and "this 'CFS-attenuation-by-inattention'". Perhaps being consistent there helps understanding: "CFS is attenuated whenever attention is directed at..." and "this 'CFS-attenuation-by-inattention'...".

5. I appreciate the openly available data and analysis scripts. In script "CFSatt_prime_bheav_analysis_main_16082022.R" on line 405, the data set "CFSlocation_data_Email.csv" is loaded but I cannot find it in the repository. Can you add this or update the scripts?

6. I recommend adding references regarding the claim that attention is typically assumed to facilitate information processing (line 99).

7. The authors excluded one participant based on high visibility (lines 285–287) even though they warn of the problems of data exclusion. I understand the reasoning for this to be that proponents of unconscious processing may reject the analysis because of a participant who indicated elevated levels of awareness. Could you quickly comment in the main text on how results change if you included this participant? I assume there are no qualitative differences, which could then be stated there. If there are deviations, they should be discussed shortly.

8. On lines 363 and 396, the authors use the label "CFA-attenuation-by-inattention" MODEL instead of, as everywhere else, HYPOTHESIS. Since they refer to a prediction of a single effect, I would stick with HYPOTHESIS unless there is a good reason to intermix the two terms.

9. On line 247, the parentheses in the reference to Ludwig et al. (2013) are off.

- Sascha Meyen (Signed Review)

Reviewer 3 ·

Basic reporting

Handschack, Rothkirch, Sterzer, and Hesselmann present an interesting experimental study on the role of selective visual attention for semantic priming under continuous flash suppression (CFS). They build on the surprising finding of a previous study by Eo et al. (2016), who reported that withdrawing spatial visual attention from the location of a CF-suppressed stimulus leads to stronger semantic processing (measured via N400 effects in ERPs). In the current behavioral priming study, the authors investigate this "CFS-attenuation-by-inattention" hypothesis, and follow up on two previous studies of their own which so far did not provide evidence for this hypothesis. The current study investigates the effect further, this time using arrow cues and peripherally presented primes that were either CF-suppressed or not. The authors test the hypothesis that attentional withdrawal promotes semantic processing and results in behavioral priming effects in a semantic number decision task. Similar to their previous results, the current also did not substantiate the “CFS-attenuation-by-inattention” hypothesis and rather find inverted priming effects in the absence of CFS.

Overall, this is a very well-written manuscript that investigates an interesting question and is relevant given the ongoing debate on the extent of unconscious processing, especially under CFS. The authors have competently implemented the study with methodological rigor and describe their procedures in detail. They also shared their data, analysis scripts and experimental code to facilitate replications or follow-up studies. I have a few comments which might help to improve some aspects of the manuscripts - but overall, in my judgment, the manuscript meets the editorial criteria and fulfills the expected standards for basic reporting, validity of the findings, and experimental design, and is thus well suited for publication in PeerJ.

Main points

(I) In the RT analysis, the authors mention that only trials with correct responses were evaluated (overall 97% of the trials). Perhaps the relatively low average error rate could be related to the rather long average RTs, e.g. if participants used a rather conservative response strategy to avoid errors. However, I wondered whether it could be useful to look at the error rates in a bit more detail and analyze them, for example, depending on CFS/noCFS conditions or attented/unattended conditions, as priming effects can sometimes also manifest in error rates. Moreover, it could be even be interesting to analyze (or plot) the error rates as a function of response time. For example, in a CFS priming study, Baumann & Valuch, 2022 (Consciousness & Cognition) recently divided the trials into 4 bins depending on RT. They found that classical priming effects tended to show up in the error rates of trials with short RTs and inverse priming effects in the error rates of trials with long RTs. They speculated that response inhibition processes could play a role and level out overall RT priming effects when both response priming and response inhibition processes are at work. The negative priming effects in noCFS trials might thus result from response inhibition effects, and it might be possible to reveal these via a fine-grained analysis of RT-dependent error rates. This type of analysis can sometimes reveal interesting additional aspects of RT data that might otherwise go unnoticed (see Panis et al., 2020; Attention, Perception & Psychophysics). I would leave it to the authors' judgment whether this makes sense in the current study (it might depend on whether or not there is sufficient data/variability in the error rates).

Experimental design

(II) In Figure 3 the results pattern seems to suggest an interaction of the factors CFS/noCFS and congruency - i.e., the congruence effects go in the opposite direction for noCFS trials compared to CFS trials. In the results section, the authors report only separate analyses for CFS vs no-CFS trials, where this specific interaction is not statistically tested. Would it be useful to test this potential interaction effect in a joint (3-way) ANOVA? If I understood it correctly, CFS and noCFS trials were mixed in the same task block, so that a joint analysis might not be completely wrong?

(III) The description of the design could specify more explicitly the proportion of trials in which the cues pointed to the task-relevant prime (the digit prime) (in the Eo study, by the way, I couldn't find this information in an explicit way either). I assume this was 50%, but I was not entirely sure about it – apologies, if I missed it. I thought about this because participants may not have much incentive to orient their attention according to the cues (an aspect already mentioned briefly in the discussion). Although one could assume that the secondary task (reporting the visibility/tilt of the attended stimulus) creates a certain incentive, I wondered whether this is sufficient. As far as I understood, the participants did not get any feedback on the task in which the orientation of the attended stimuli was to be reported (and naturally also not on the alternative task of the PAS rating) - therefore, it is hard to tell in retrospect whether the participants reported the cued or the non-cued stimulus (or maybe switched between the two, depending on what they saw) - and this could be mentioned as a limitation more clearly. In classical cueing paradigms (e.g., Posner, 1980), with central/endogenous cues, unequal ratios of valid vs. invalid cues are often used to promote a strategic use of the cues, e.g., in 66% of trials the cue would appear at the location of the subsequent task-relevant stimulus so that participants actually pay attention to the cues and orient their attention accordingly. Of course, this is not easily translatable into the present setup, where the targets are presented centrally, but since participants were asked to judge the visibility of the primes, a predictive cue might have a stronger effect and result in larger attentional effects if the secondary task would have had a clearly correct or wrong response on which they could have received feedback. Another possibility would be to present peripheral (exogenous) cues at the positions of the peripheral prime stimuli, which could also elicit more reliable effects on attentional orienting even in the absence of cue predictivity. This is not a strong criticism of the present study as it is difficult to know which kind of procedure would work best, and I mean it more as a suggestion for potential follow-up studies.

Validity of the findings

The findings are valid.

Additional comments

Minor points

(L188) Were the cues also presented in the staircase procedure (same as in the main experiment), and were the stimuli presented at the same peripheral positions. Also, what exactly was the task (and the instructions)? A few more details would be helpful here.

(L212ff) In the methods section, the prime digits (1, 3, 5, 7) - see also line 232, are probably a bit mixed up. From the context of the task and what is mentioned in the results section, I assume that these should be (1, 3, 7, 9) instead. You could also briefly mention why the digits/letters were randomly tilted (it becomes clear that this is necessary for the secondary task later, but naïve readers would need to piece this together themselves).

(L232) The mention of the individual prime digits is redundant with the information at the beginning of the section (L212).

(L238) I wondered why fewer trials were recorded for participant #3 and what makes this participant so special. If this participant was excluded from the analyses anyway, maybe it would suffice to briefly mention the reason for exclusion in the methods section and then omit bringing up this excluded participant repeatedly in the methods/results? After all, the other excluded participants are also not discussed in more detail (or included in the presentation of results) so is there something particular about this participant that it deserves special discussion?

(L382) Since a considerable part of the results section is dedicated to the analysis of possible response conflict effects, these analyses should also be at least briefly motivated in the introduction - some readers might initially read only the introduction and discussion, and then wonder what the authors are talking about when it comes up in the discussion.

(L484) The authors write "In the second control experiment, we tested whether the low-contrast digits and letters were fully visible when no CFS masks were presented." – Is this merely an imprecise expression or was this actually a deviation from the main experiment’s procedure (and if so, could you briefly mention the reason?). As far as I understood, in the main experiment, CFS masks were also presented in noCFS trials (except that there was no interocular suppression).

(L489) "Next, the subjective visibility of the attended stimulus had to be rated. The mean PAS rating was 3.80 ± 0.07 SEM." If I understood it correctly, this means that in the visibility tasks either the relevant (digits) or the irrelevant (letters) stimuli should be reported (and this was also the case in the main experiment) - correct? If so, it might be helpful to mention this more explicitly in the methods section and mention that the idea was to foster attentional orienting to the cued position – irrespective of whether it contains the digit or the letter stimulus (e.g., line 223) – I spent some time thinking about this because it is different from some other priming studies that often assess the visibility of the stimulus that should be responsible for the priming effect (which was still done in the separate control experiment in the present study).

---

## Round 0.2 · Minor Revisions

First of all, I want to thank the Reviewers for their time and efforts in revising this new version. I really appreciated their timely responses.

As you will see, all three reviewers are happy with this new version and I can only share their enthusiasm. Two reviewers endorsed the publication. One reviewer proposed some further, small, changes: these concern the reported analyses (plus other minor suggestions). I, therefore, would invite you to provide a new version of the manuscript, in order to swiftly proceed to the final decision.

·

Basic reporting

see review

Experimental design

see review

Validity of the findings

see review

Additional comments

Review of Ms. #76930, "The scope and limits of numerical priming under interocular suppression – No effect of attentional modulation by spatial cueing", by Juliane Handschack, Marcus Rothkirch, Philipp Sterzer, and Guido Hesselmann

Reviewer: Thomas Schmidt

My main concerns had been whether or not the task at hand is indeed semantic, whether there were large variations in stimulus conditions across observers, and whether the dual task demands lead to overly slow response times. I also asked for more information about the Bayesian priors. The authors have carefully addressed all my concerns and now present a very fine paper. I recommend publication as is.

·

Basic reporting

The manuscript was written in a clear and understandable way and has substantially improved further in clarity since the first submission. The relevant literature is discussed and the authors have added additional literature where small gaps were pointed out. Figures are improved to show the relevant effect sizes. The structure of the manuscript is such that I do not find anything to improve.

Experimental design

The research question is well defined and I stand corrected by the authors: The manuscript does fit well into a group of previously published papers in the journal. The methods are sufficiently detailed to allow for a replication. The authors improved reporting of measurement errors such that I have only one easy-to-do remaining issue regarding the reported analyses.

Validity of the findings

The motivation for the study is solid. In their revised manuscript, the authors improved the fit between data and interpretation to the degree that I am satisfied with. The authors now share additional data and completely fulfill the open data policy.

Additional comments

MAJOR ISSUES

1. DISCUSS THE MISSING PRIMING EFFECT IN CFS TRIALS

The authors made the changes I wanted to see. I have one more small suggestion: Report a confidence interval or SEM for the attenuation effect in the CFS condition. To motivate this suggestion, I want to elaborate on the issue to foster understanding. But, to be clear, this discussion should not hinder publication and my other issues were also resolved.

In my first review, I pointed to the missing priming effect in CFS (invisible) trials. I agree with the authors that, since Eo et al. (2016) did not report a priming effect, this need not be framed as "failure to replicate". But the main problem remains: It is questionable to investigate the modulation of an effect if the effect is not there in the first place. Crucially, one would expect an effect in the CFS condition because this is the only relevant condition for the 'CFS-attenuation-by-inattention' hypothesis.

Eo et al. (2016) state in their discussion: "we provide evidence that attention modulates the extent of unconscious semantic processing of INVISIBLE STIMULI DURING CFS" (p. 5495, emphasis added). Thus, Eo et al. start with an effect (in N400) in the CFS trials and then find it to be modulated by inattention. Now, the current manuscript investigates the hypothesis via RTs (instead of N400, which is fine) but they don't find the effect in RTs in CFS trials, which should have been the starting point. The authors replied: "We agree: the starting point when investigating the 'CFS-attenuation-by-inattention' hypothesis should be a paradigm in which participants make use of the information conveyed by the prime [...] [which] is the case in our study too, because we observe a significant effect of the primes on RTs in no-CFS trials [...]". But the no-CFS (visible) trials are irrelevant to the CFS-attenuation-by-inattention hypothesis.

Nevertheless, the results provided in this manuscript are an important contribution and the authors diligently implemented my suggestion so that I now consider their manuscript to be self-contained. (For example, I am happy the authors now discuss their visibility results in lines 598-599 which suggest that the primes were bordering on being too visible so that it is implausible that the priming effect in the CFS trials could be increased without making the prime stimuli visible.) Still, it is important that the interpretations regarding the CFS-attenuation-by-inattention effect remain as they are now (ll. 44–45, 376–377, and 540–541) and don't become stronger.

To facilitate future research discussions, I would like to ask for a confidence interval or SEMs for the attenuation effect under CFS. As far as I see now, the observed priming effects in the CFS trials are 7 ms (unattended) vs. 10 ms (attended). Thus, the observed CFS-attenuation-by-inattention effect is 7 - 10 = -3 ms and I would like to see the appropriate confidence interval or SEM for this value either in figure or text.

2. EFFECTIVE DESCRIPTIONS

The authors addressed this issue to my full satisfaction. I enjoyed reading the improved structure of the manuscript (and Kruschke HDIs were only an optional suggestion).

MINOR ISSUES

Almost all minor issues were fully addressed. Two remaining, very minor things:

1. In lines 85-86, the reference to Lin and He (2009) is a bit awkwardly formatted with the square brackets.

2. In line 266, the reference to Ludwig et al. (2013) still needs a little change. Right now there is one open parenthesis too many.

For the record, I really liked "not all invisible stimuli are equally invisible" too.

- Sascha Meyen (Signed Review)

Reviewer 3 ·

Basic reporting

no comment

Experimental design

no comment

Validity of the findings

no comment

Additional comments

I thank the authors for considering my suggestions. I have now read the revised version in detail. All my previous comments were very well addressed by the authors and I have no reservations about publishing it. I also believe the current paper fits very well into PeerJ and connects to previous publications on the topic of unconscious visual processing that have been published here.

---

## Round 0.3 · accepted · Accept

I have assessed the revision myself, and I am happy with the current version. I feel that now the paper is ready for publication. Many thanks for your hard work and congratulations!